# Automatic Rural Road Centerline Detection and Extraction from Aerial Images for a Forest Fire Decision Support System

**Miguel Lourenço** [1], **Diogo Estima** [1], **Henrique Oliveira** [2,3,*], **Luís Oliveira** [1,4] and **André Mora** [1,4]

1 Department of Electrical and Computer Engineering, NOVA School of Science and Technology (FCT NOVA), NOVA University Lisbon, 2825-149 Caparica, Portugal
2 Telecommunications Institute, 1049-001 Lisbon, Portugal
3 Polytechnic Institute of Beja, 7800-295 Beja, Portugal
4 Centre for Technologies and Systems (UNINOVA-CTS), 2829-516 Caparica, Portugal
* Correspondence: hjmo@lx.it.pt

**Abstract:** To effectively manage the terrestrial firefighting fleet in a forest fire scenario, namely, to optimize its displacement in the field, it is crucial to have a well-structured and accurate mapping of rural roads. The landscape's complexity, mainly due to severe shadows cast by the wild vegetation and trees, makes it challenging to extract rural roads based on processing aerial or satellite images, leading to heterogeneous results. This article proposes a method to improve the automatic detection of rural roads and the extraction of their centerlines from aerial images. This method has two main stages: (i) the use of a deep learning model (DeepLabV3+) for predicting rural road segments; (ii) an optimization strategy to improve the connections between predicted rural road segments, followed by a morphological approach to extract the rural road centerlines using thinning algorithms, such as those proposed by Zhang–Suen and Guo–Hall. After completing these two stages, the proposed method automatically detected and extracted rural road centerlines from complex rural environments. This is useful for developing real-time mapping applications.

**Keywords:** rural roads; centerline extraction; deep learning; geographic information system (GIS); wireless sensor networks (WSN); decision support system (DSS); convolutional neural network (CNN); spatial pyramid pooling (SPP); forest fires





## 1. Introduction

Various regions in the world are frequently affected by forest fires, including North America, Australia, and the southern countries of Europe. In Portugal, a country in southern Europe, forest fires are problematic and increasingly frequent, with significant ecological and socio-economic impacts. In the last 20 years, the country has been severely affected by large forest fires, which have destroyed the environment, damaged residences, and claimed lives. The most extensive burned area in the last ten years happened in 2017, with over 21,000 forest fires burning over 500,000 hectares of forest and taking over 114 human lives.

Portugal is a country affected by high temperatures during the summer season, along with strong winds coming from the Atlantic, with forest fires powered by multiple factors, namely: (i) the climate, causing a substantial increase in biomass volume; (ii) the decrease in agriculture and pastoralism activities, causing an increase in biomass in some areas; (iii) the lack of land use planning, resulting in more profitable crops and leading to more fuel for forest fires; (iv) the fact that 92% of the forest land is owned by private owners, while the state owns only 3%, with about 6% owned by local communities, a scenario that causes difficulties in enforcing forest planning laws [1]. In addition, many rural areas of the country are becoming increasingly depopulated. Since private owners control the vast majority of the forest land, with the lack of forest management, trees, dead leaves, bushes, grasses, and fallen pine needles quickly accumulate, acting as fuel for forest fires. Due to

climate change, more frequent and extreme forest fire events are expected to occur in the coming years.

Firefighters must reach the epicenter of a forest fire as quickly as possible, and rural roads are often used for this purpose. Typical road navigation applications only provide access to primary and secondary roads, lacking information regarding tertiary/rural roads. Forest fires are typically located in areas with very rough topography that can only be reached by tertiary/rural roads, therefore, it is essential to correctly map them. A road network is a valuable data source for terrestrial fleet management to guide firefighters in a forest fire scenario. The availability of a suitable rural road network enables the development of many valuable functionalities that can help firefighters in a forest fire scenario, such as: finding (i) the closest path to a forest fire spot; (ii) the fastest emergency routes; (iii) dead-end roads; or even (iv) the shortest path passing through multiple locations to reach a specific forest fire spot.

A road network is an essential data source used in some Geographic Information Systems (GIS) applications, such as car navigation systems, and is composed of various interconnected line segments (vector data) representing the geographic center of road pavement surfaces. However, building road networks in digital format is a significant effort that needs to be undertaken at the regional level and a massive one at the national level. To avoid manually mapping extensive road networks, an automated method is required to generate them based on the digital processing of aerial or satellite images. Moreover, the landscape complexity, the existence of multiple road materials (e.g., dirt, asphalt, cement, and gravel), and road occlusions [2] cast by vegetation (like trees, small bushes, and dense vegetation) make it challenging to extract completely smooth, and accurate road centerlines, which demands sophisticated methods to process those aerial and satellite images. This is the motivation to present an efficient and precise method to automatically detect and extract road centerlines from aerial images that include rural roads.

To achieve this goal, this article proposes an automated methodology capable of: (1) detecting rural roads from aerial images; (2) and extracting their centerlines. The whole process will be divided in two major stages: road detection and road centerline extraction.

The road detection process must be accurate enough to ensure that all the road pixels are detected in the aerial images. To achieve this, the DeepLabV3+ architecture was used to detect rural roads and predict a binary mask of roads and background elements. With all of these pixels together we completed the road detection step.

Road centerlines are vector line data that represent the geographic center of road rights-of-way on transportation networks. The road centerlines will be extracted from the previous road detection. In the second stage, an algorithm is applied, to optimize road connectivity and remove possible artifacts (unwanted objects) that might wrongly appear in the prediction phase. Finally, the road centerline from the previously detected roads is extracted using a skeletonization method.

With the proposed architecture, it is expected to obtain a precise and clear rural road centerline that should be a white one-pixel-wide object, presented in a black background. It is not only essential to detect rural roads but extract their centerlines to produce vector line data capable of being used to build network datasets for powering routing applications. This article is structured as follows. After this introduction, Section 2 presents related works on state-of-art road detection and road extraction algorithms or methods from aerial images. Sections 3 and 4 describe the proposed method and its implementation, respectively, while Section 5 presents the experimental results on predicting roads, including a description of the evaluation metrics used, the results of rural roads centerline extraction, and some additional optimizations. The article ends with a discussion of the results (Section 6), conclusions and future work (Section 7).

## 2. Related Work

Researchers and national authorities have considered solutions to minimize the burning of forests, mainly based on the different stages of a forest fire. Regarding prevention,

some solutions have been proposed, mainly: (i) optimizing land-use planning or making firebreaks along the forest to minimize the speed fire spread [3]; (ii) the use of imaging sensors to detect forest fires at an early stage, such as the installation of powerful vision or infrared cameras at critical spots [4]; (ii) wireless sensors networks (WSN) that can predict forest fires by detecting the change in some environmental parameters, such as the decrease of humidity and the increase of toxic gases [5].

Other solutions address the planning and support actions regarding the suppression operations of a forest fire. One such solution is the Decision Support System (DSS), developed by Portuguese researchers [6]. It is based on mobile devices and can be used real-time during a forest fire, overlaying several data layers in a GIS environment. It is essential to provide commanders with decision support systems that use accurate, updated and concise data to facilitate decision-making. When possible, new tools should be fostered to support this process [7]. One of the fundamental GIS data layers that need to be included in the system is the rural road network of a potential forest fire area. These specific data are beyond the typical urban setting and is missing from in-car navigation maps. It can be essential to access the rural road network since most forest fires cannot be accessed through the conventional road network. In this article, an automated method is proposed to automatically detect and extract the rural road network, to use in in-car navigation.

One way to build a digital mapping of roads to be included in a GIS environment is to extract their pavement centerlines in a vector data format after detecting them from aerial image processing. The use of convolutional neural networks (CNN) to perform road detection and centerline extraction has increased in the last few years. These deep learning methods have achieved better results than other methodologies and they can handle complex landscape images with high accuracy.

The variety of land objects visible in images can have a significant impact on the learning algorithm used to detect rural roads, in particular: (i) different types of roads with pavements of various colors; (ii) different types of trees may also have various colors; (iii) road occlusions due to shadows cast by vegetation; (iv) light variations due to solar attitude, among other issues. This image data variability brings significant challenges for example: (i) the grey pixels of a road might be hidden by a black shadow cast by a tree; (ii) the shape of a road might be distorted due to its occlusion; (iii) the color of a road outline may be very similar to the landscape, becoming very hard to from the surrounding pixels. Deep learning methods can help to overcome these problems and protect the learning algorithm from these high-level abstractions.

When it comes to road detection algorithms, many CNN-based architectures have been proposed in the past years, namely: (i) CasNet, which has shown a lot of visual and quantitative advantages compared to other state-of-the-art methods, but it has a higher level of difficulty associated with its implementation [8]; (ii) FCNs, which has shown very significant results in road connectivity, accuracy and completeness [9]; (iii) DCNNs [10], which can allow overcoming the problem of complex image backgrounds, effectively overcoming the phenomenon of "burr" with high computational speed and accuracy, although further improvements are necessary to reduce the impact of shadows, trees, and buildings [11]. (iv) U-nets, that enables precise pixel location since U-net uses skip connections to associate low-level feature maps with high level feature maps. It allows for fast predictions with a simple architecture [12].

Regarding the road extraction phase, many solutions were analyzed, namely: (i) in [13], a self-supervised learning framework was proposed to automatically extract road centerlines. This approach does not require to manually select the training samples and other optimization processes. It achieves better results quantitatively and visually, compared with the traditional supervised road extraction algorithms, and achieves superior noise resistance than previous unsupervised algorithms; (ii) in [14], an architecture for semantic pixel extraction named SegNet was presented. This solution has achieved practical trade-offs in terms of balancing the training time, and memory versus accuracy.

Based on this, deep learning-based methods were chosen in the system described in this article, aiming at improving road detection and center-line extraction considering the existence of obstacles and/or different road materials.

## 3. Proposed Architecture

The proposed method uses the state-of-the-art DeepLabV3+ architecture and thinning algorithms to detect and extract rural road centerlines from aerial images. The main reasons why this architecture was chosen were: (i) it achieved a significant performance on cityscapes datasets without any post-processing, with the PASCAL VOC 2012 model becoming a new state-of-the-art (ii) since rural roads are made of many types of road materials such as dirt, asphalt, gravel, and cement, an architecture based on deep convolutional neural network is expected to be able to detect them [11]; (iii) it takes advantage of the Spatial Pyramid Pooling (SPP), and the Encoder-Decoder architecture, which can be extremely useful as aerial rural images have very challenging environments with complex landscapes, frequent road occlusions, etc. [15–17].

Regarding the SPP and the Encoder-Decoder architecture:

- The SPP [18–20] encodes multi-scale contextual information, which means that it gives the network the ability to extract knowledge or apply knowledge to the information and does not require a fixed-size input image;
- The Encoder-Decoder architecture has been proven to be very useful in image segmentation [8–10,21]. The encoder progressively diminishes the feature maps and acquires high semantic information, while the decoder progressively recovers the spatial information. The Encoder-Decoder can extract sharp object boundaries, and it also helps to extract features by using atrous convolution.

Figure 1 represents the workflow of the proposed method, including road detection (the first stage) and road centerline extraction (the second stage). In the first stage, the aerial images are processed by the DeepLabV3+ model to learn to detect rural roads and predict a binary mask of roads and background elements. For the training, validation, and testing of the CNN road detector, a dataset of aerial images and a binary mask of detected roads (ground truth) were prepared and included. In the second stage, an algorithm to optimize road connectivity and remove possible objects that might wrongly appear in the prediction phase is applied. Finally, the road centerline from the previously segmented roads is extracted using a skeletonization method. The output is an image with clean white road centerlines on a black background.

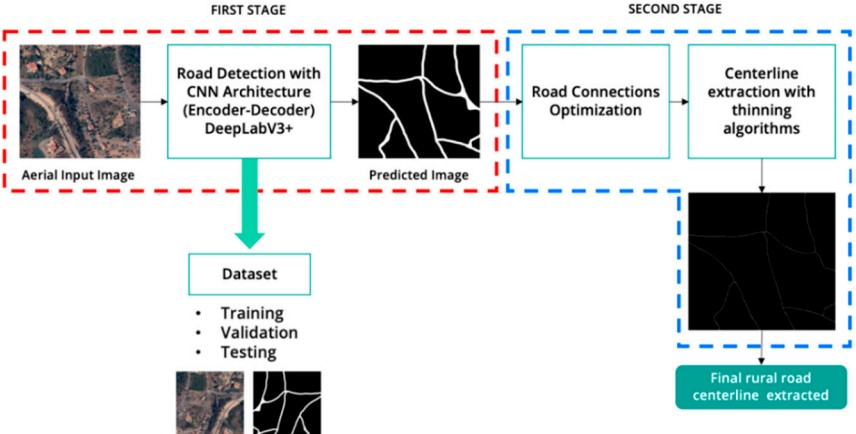

**Figure 1.** Workflow of the proposed method, including road detection (first stage) and road centerline extraction (second stage).

For the extraction of one-pixel-wide road centerlines, various thinning algorithms were tested, including some of the most widely used developed by Zhang–Suen [22] and Guo-hall [23]. A thinning algorithm aims to take a binary image, with pixels of road regions

extracted labeled with "1" and the background pixels labeled with "0", to draw a one-pixel wide skeleton on the processed image while maintaining the shape and structure of the road. Thinning algorithms are one of the most practical ways to extract the road centerlines, even though it sometimes produces small spurs around the centerline that can affect the final structure of the road network.

With the proposed architecture, it is expected to obtain a precise and clear rural road centerline. In the end, the rural road centerline should be a white object one-pixel-wide, presented in a black image background.

## 4. Methods

### 4.1. Dataset Preparation

The municipality of Mação, located at the center of mainland Portugal, was the study area chosen for this article (see Figure 2) because it has been severely affected by forest fires in the past [24].

The municipality of Mação with its road network

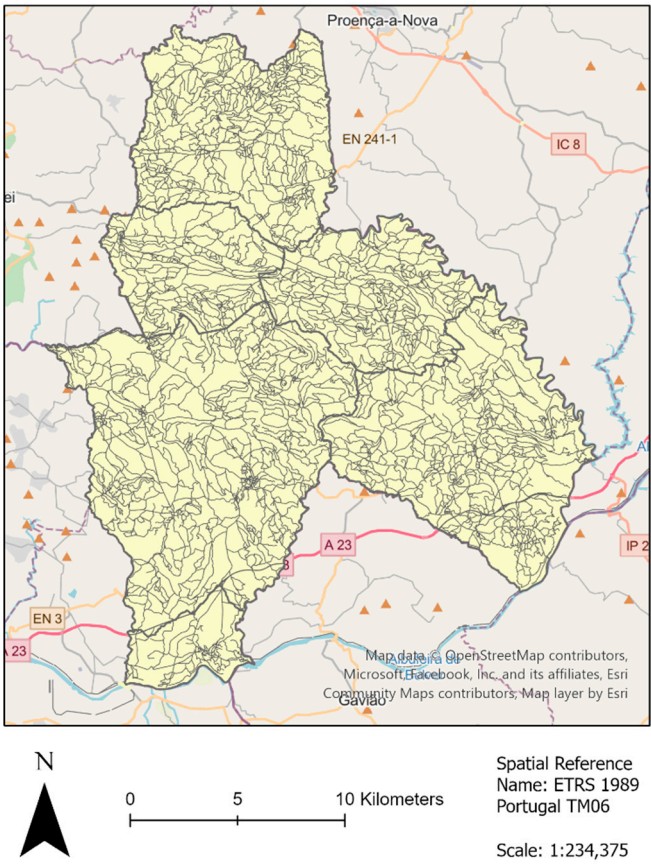

**Figure 2.** The road network of the Mação municipality, with an area of approximately 400 km$^2$.

Sixty-one high-resolution orthorectified aerial images provided by the Direção Geral do Território (DGT—[25]) were used in this research. They were in Tagged Image File Format (TIFF), with an intensity resolution of 8 bits in RGBI (Red, Green, Blue, and Infrared channels) and a spatial resolution of 0.25 m. Only the RGB channels were used, with each aerial image presenting 16,000 × 10,000 pixels covering an area of 4 km by 2.5 km.

The associated ground truth images were created manually, overlaying a black layer on top of the aerial image and highlighting the rural roads in white. This process applied several aerial images, creating multiple pairs of aerial images with their corresponding

masks. The next step was to cut the large aerial images (16,000 × 10,000 pixels) into smaller tiles (1024 × 1024 pixels) (see Figure 3), obtaining a total of 486 images.

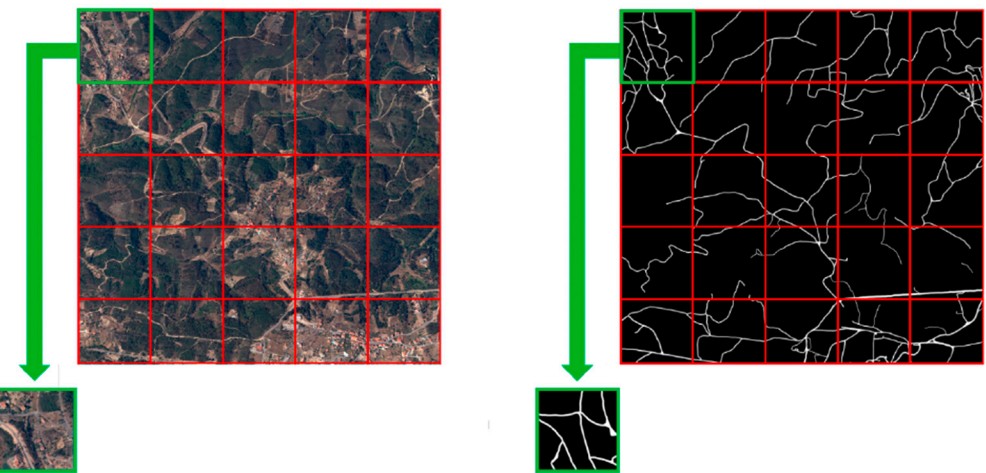

**Figure 3.** Original orthorectified aerial images (**left**) and the corresponding masks of size 1024 × 1024 pixels (**right**), each tile covering an area of 256 × 256 m$^2$.

Creating the ground truth masks by manually outlining the rural roads is a very tedious and time-consuming task that limits the size of the initial dataset. Thus, a data augmentation procedure was used to enlarge the dataset. The processes started by rotating ninety degrees four times the initial 486 orthorectified aerial image tiles of size 1024 × 1024 pixels, increasing the dataset size to 1944 images. Then, a mirroring operation was performed, resulting in a total of 3888 images (see Figure 4).

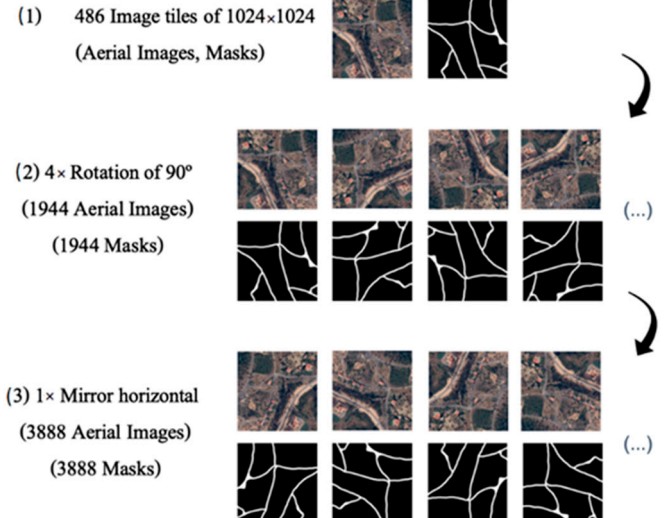

**Figure 4.** Enlargement of the dataset with a data augmentation procedure.

### 4.2. Road Detection Classifier Definition

In this article, the proposed method uses the state-of-the-art deep learning model for semantic image segmentation—the DeepLabV3+ by Google—to perform rural road region detection [26]. DeepLabV3+ has strong architectural characteristics to allow solving this task, such as:

- **Spatial pyramid pooling**: has been proven to be a flexible solution for handling different scales, sizes, and aspect ratios [17,27]. Moreover, it encodes multi-scale contextual information by probing the incoming features with filters and applying several parallel atrous convolutions with different rates. This means that atrous

convolutions can control the resolution of how features are computed, also known as Atrous Spatial Pyramid Pooling (ASPP). Atrous convolution grants us the ability to control the feature maps resolution inside the model and adapt the filter's field of view to capture information without a specific size. Considering the feature map output as $Y$, the convolution kernel as $w$, and the input feature map as $x$, the atrous rate $r$, for each location $i$, the convolution operation is given by Equation (1):

$$Y[i] = \sum_k x[i + r.k].w[k];$$   (1)

- **Encoder-Decoder**: its architecture has been widely used in semantic segmentation [14,28,29], consisting of two main parts. The encoder will progressively reduce the spatial size of the feature maps and gather high semantic information, while the decoder will recover the spatial size and detailed object boundaries. This means that the structure can extract sharper object boundaries by progressively recovering spatial information;
- **Depthwise Separable Convolution**: has been applied in recent neural network designs [30]. Moreover, it has the primary purpose of dramatically reducing the overall computational costs and number of parameters while keeping an equal or even higher performance. This result is achieved by performing depthwise spatial convolution independently for each channel, followed by a pointwise convolution ($1 \times 1$ convolution).

For road region detection, DeepLabV3+ was used as an Encoder-Decoder architecture (see Figure 5). The encoder takes the input image, and the decoder outputs the binary mask. For the network backbone, the ResNet50 was chosen, using the pre-trained weights of *imagenet*, allowing for faster training of the model. The output of ResNet50 goes into the ASPP module, where a $1 \times 1$ convolution is performed to reduce the computation cost and the number of parameters being used. Then, $3 \times 3$ convolutions with rates of 12, 24, and 36 are executed, followed by image pooling (average) to extract the features, and reduce the image size. After that, all the layers are concatenated, and another $1 \times 1$ convolution is applied to reduce computational costs, upsampling the features by 4. Regarding the decoder part, the processing starts with a $1 \times 1$ convolution, followed by the concatenation using the feature map from the encoder, where another $3 \times 3$ convolution is applied, upsampling the features by four so that the output has the same resolution as the input image.

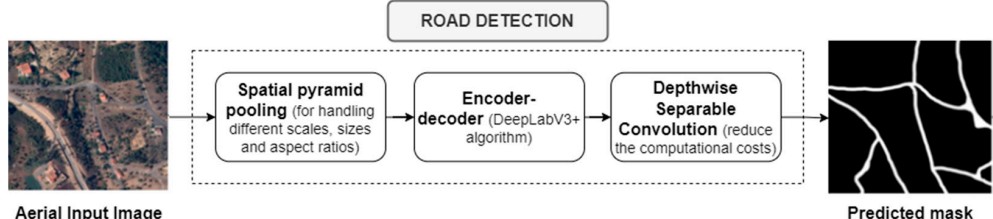

**Figure 5.** Road region detection module of the rural road centerline extraction method.

A Python image segmentation neural network framework based on PyTorch was used to implement the proposed DeepLabV3+ model. The model parameters used to build the proposed road detection architecture are as follows:

- *encoder_name*: this parameter represents the feature extractor encoder, commonly known as the network backbone, extracting features with different spatial resolutions from the input image. For the backbone, a Residual Neural Network (ResNet) is used. The chosen residual network was ResNet50 since it is a convolutional neural network with 50 layers (48 Convolution layers, 1 Max- Pool, and 1 Average Pool layer). ResNet uses skip connections to add the output from a previous layer to further layers, helping mitigate the vanishing gradient problem, providing more depth, and reducing computational resources;

- *encoder_depth*: corresponds to the number of downsampling operations inside the encoder. It can vary between 3 to 5. In this article, the encoder depth considered was equal to 5. In each stage, the feature map size decreases by half compared to the previous stage;
- *encoder_weights*: represents the multiplication factor of the kernels in the convolutional layers. The *imagenet* [31] is used as the pre-trained encoder weights. Using a pre-trained encoder vastly speeds up the training time of the model;
- *encoder_output_stride*: represents the relationship between the input and output image resolutions of the last encoder features. An *encoder_output_stride* = 16 is used for the best relationship between speed and accuracy;
- *decoder_atrous_rates*: are the dilatation rates for the ASPP unit. *Decoder_atrous_rates* = (12, 24, 36) are used;
- *classes*: represents how many classes the output has. Rural region road detection only implies two classes: road and background;
- *activation*: is the function used next to the last convolutional layer to generate the output based on the inputs. Some examples of activation functions are **sigmoid**, **tanh**, **logsoftmax**, **softmax**, and **identity**. Since the detection problem only has two classes (road, background), it is recommended to use a two-class logistic regression by using the sigmoid activation function;
- *upsampling*: is the factor that keeps the same input-to-output ratio. This factor is equal to 4 since, after the encoding module, the feature sizes are decreased by 16;
- *epochs*: corresponds to the number of complete passes by the algorithm in the entire training dataset. Three training epochs are set throughout the dataset;
- *loss*: is used to measure how well a prediction was made, comparing the predicted value with the ground truth value. In this case, the *DiceLoss* function is used, which is one minus the dice coefficient;
- *metrics*: measures and monitors the model's performance during the training and testing phase. The **Jaccard** index was initially used, but state-of-the-art metrics were adopted later;
- *optimizer*: **Adam** is used for the optimizer, an algorithm for first-order gradient-based optimization of stochastic objective functions [32]. This method requires less memory, is computationally efficient, and is appropriate for problems with multiple parameters and data. The hyperparameters also have an intuitive understanding, requiring almost no adjustment. The optimizer is set with a learning rate of $8 \times 10^{-5}$.

### 4.3. Road Detection Classifier Training

To make predictions, deep learning models need to learn the mapping relationships between inputs and outputs. This process includes discovering a series of weights that are a good fit to solve a specific problem, which in the scope of this article, is to teach the model to distinguish between rural roads and background pixels. After creating the training epochs, the model can be trained. The training process consists of simply looping through the data iterator, supplying the inputs to the network, and optimizing it. After iterating through the number of epochs, the model is saved as a *.pth* model. This process took approximately 3–4 h per training using Google Colab GPUs.

### 4.4. Road Centerline Extraction

The DeepLabV3+ model presented some difficulties regarding the intersection of roads made of different types of materials (i.e., asphalt roads and dirt roads, among others). Therefore, the road detection optimization step presented in this section proposes an approach to overcome these difficulties and remove small white objects (artifacts) wrongly predicted as roads. As shown in Figure 6, it is a four-stage process aiming to detect rural roads' centerlines with a single pixel width. First, the algorithm improves the connections of the rural road intersections and removes noise from the predicted image.

Then, morphological image thinning algorithms are applied to extract road centerlines, having tested two of the most well-known methods: Zhang–Suen; and Guo–Hall.

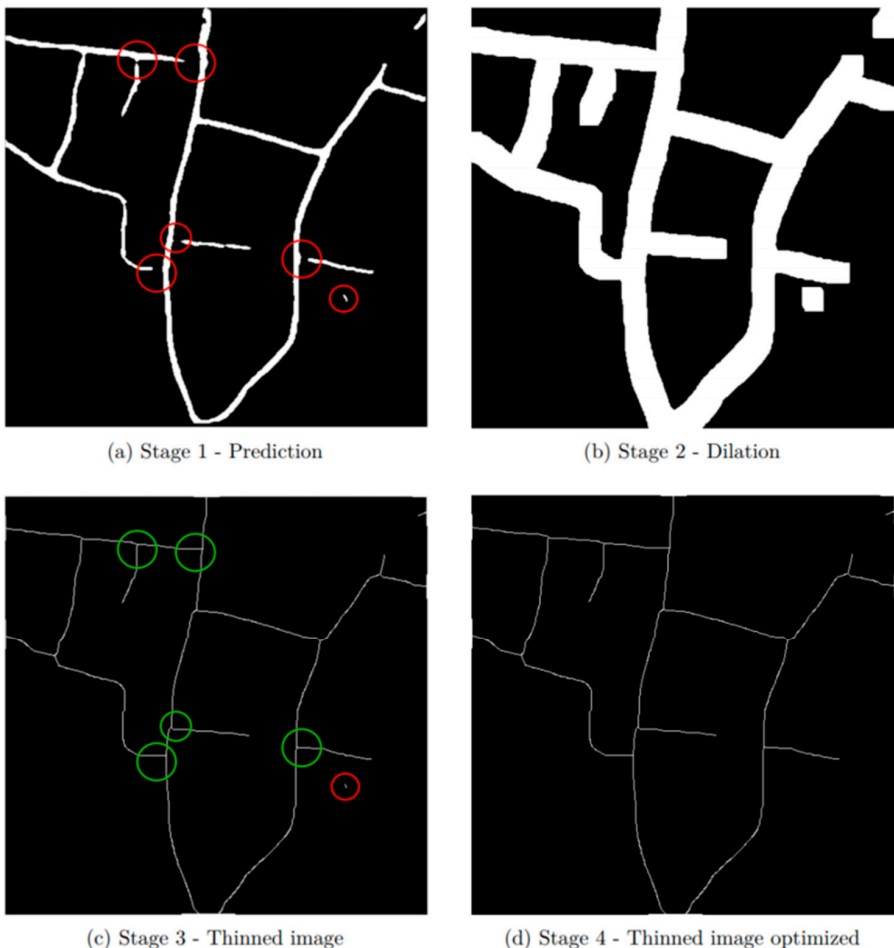

**Figure 6.** Sample results of road connection optimization process: (**a**) stage 1—prediction (red circles represent missing intersections and an artifact); (**b**) stage 2—dilation; (**c**) stage 3—thinning (the intersections in green circles have been corrected; the artifact in the red circle is going to be removed); (**d**) final results after removing smaller objects.

The thinning algorithm is used to take a binary image (in our case, a white road in a black background) and draw a one-pixel-wide skeleton of the white objects while maintaining the shape and structure of the road. These algorithms are one of the simplest ways to implement road extraction. However, they often produce minor artifacts around the centerline that can affect the final structure of the road network.

Figure 6a shows an example of rural road (first stage) detection results, in which five incomplete road intersections were highlighted by red circles, with one short object also highlighted by a red circle. The second stage aims to connect rural roads of different types that are nearby by executing five iterations of a dilation using a $5 \times 5$ kernel window, whose results are shown in Figure 6b. In the third stage, the thinning of the previous dilated road regions is performed, resulting in correct rural road intersections, highlighted by the five green circles in Figure 6c. Finally, the removal of small objects, such as the one highlighted by the red circle in Figure 6c, is addressed. The procedure uses the exhaustive connected components methods to label all objects in the mask and remove all with a length smaller than 140 pixels (empirically chosen after exhaustive testing and only the image database used within the scope of this research work). The final results are presented in Figure 6d, which shows a well-draw rural road network with one-pixel-wide segments over a black background.

## 5. Evaluation Results

### 5.1. Evaluation Metrics

Multiple metrics were defined to evaluate the road detection model. A study of the confusion matrix was undertaken, which provides an in-depth analysis of the proposed method's performance. From the latter, four important indicators were extracted:

- **True Positives (TP)**: represents the road pixels correctly detected;
- **True Negatives (TN)**: represents the background pixels correctly detected;
- **False Positives (FP)**: represents the road pixels that were incorrectly detected;
- **False Negatives (FN)**: These represent the road pixels that were not detected.

Three benchmark metrics proposed by Wiedemann [33,34] were used to assess the quantitative performance in the road region detection and the centerline extraction process, namely: (i) *completeness* (COM) in Equation (2); (ii) *correctness* (COR) in Equation (3); (iii) *quality* (Q) in Equation (4). Another additional metric, known as *F1-score* (F1) in Equation (5), was also used [35].

$$COM = \frac{Length\ of\ matched\ reference}{Length\ of\ reference} \approx \frac{TP}{TP + FN}\epsilon[0;1] \tag{2}$$

$$COR = \frac{Length\ of\ matched\ extraction}{Length\ of\ extraction} \approx \frac{TP}{TP + FP}\epsilon[0;1] \tag{3}$$

$$Q = \frac{Length\ of\ matched\ extraction}{Length\ of\ extracted\ data + Length\ of\ unmatched\ reference} \approx \frac{TP}{TP + FN + FP}\epsilon[0;1] \tag{4}$$

$$F1 = \frac{2 \times COM \times COR}{COM + COR} \approx \frac{2 \times TP}{2 \times TP + FN + FP}\epsilon[0;1] \tag{5}$$

For road centerline extraction, the approach needs to be slightly changed. As the human operator manually extracts the roads, discrepancies may occur between the manually labeled centerline and the true centerline. This means that it is not suitable to compare the centerline extracted and the ground truth centerline with a single-pixel width, as one tiny shift of pixels can influence the whole performance of the model. Thus, a buffer method is used to solve this issue, which compares the matching extracted data with the reference data, where every section of the network is within a given buffer width $\rho$.

The reference centerline with a specific buffer width $\rho$ is dilated to find the TP and FP pixels (see Figure 7). The next step involves making the intersection between the dilated reference data and the extracted centerline data, resulting in the matched (TP) and unmatched extracted data (FP). Dilation of the extracted centerline with a buffer width $\rho$ is performed to find the FN pixels, followed by an intersection with the reference data, resulting in the matched and unmatched reference data (FN). After this procedure, it is possible to calculate the previously defined metrics in Equations (2)–(5).

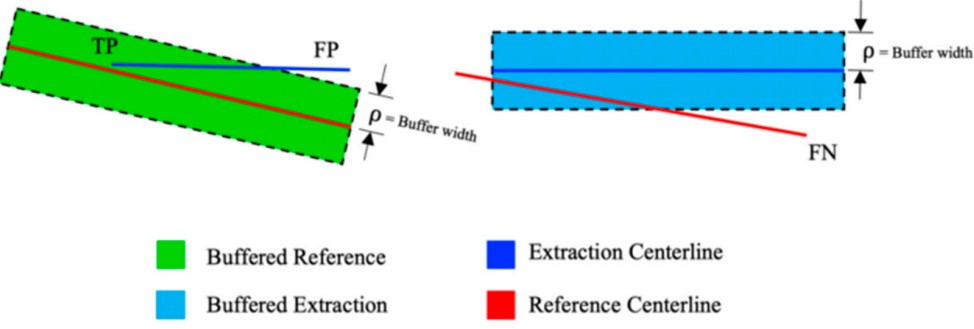

**Figure 7.** Road centerline extraction using the buffer method [33].

### 5.2. Road Detection Evaluation

A comparison with other state-of-the-art approaches addressing road detection and road centerline extraction was executed to assess the performance of the proposed architecture. Since the dataset used does not belong to a benchmark dataset because it is unique, the output results would be biased with the proposed implementation. Thus, it was decided to make a comparison as described below.

A qualitative and quantitative comparison was made between three state-of-the-art methods using deep neural networks for road detection. The first used the DeepLabV3+ model, chosen for rural road detection. The second was Unet [36], a model that has achieved a new benchmark performance and is becoming a state-of-the-art method for biomedical image segmentation. The third method was the FPN model [37], which was developed by Facebook AI Research (FAIR) and has achieved state-of-the-art single-model results on the COCO detection benchmark, outperforming all existing single-model entries, including the COCO 2016 [13] challenge winners.

All the road detection experiments were performed using the same initialization and optimization parameters defined in the training procedure. In all the tests, the same learning rate was used, as well as the number of epochs that was set to 3, the network backbone (ResNet50) used pre-trained weights (ImageNet), a batch size that was set to 4, two output classes, and the same activation function (sigmoid). The models were trained using Google Colab Pro GPUs inside a macOS operating system with a 2.6 GHz 6-core Intel Core i7 processor.

By using DeepLabV3+, it is possible to detect roads in complex rural environments. Combined with the Encoder-Decoder, the SPP allows the model to extract sharp features around the edges of road regions. This model was able to detect roads occluded even by shadows, trees, and other objects in the middle of the road, being a robust option for rural road detection. The DeepLabV3+ model had some difficulties regarding connecting different types of roads (i.e., asphalt, dirt, cement, grass), resulting in incomplete connections of rural roads. The Unet model allowed us to obtain more complete connections than DeepLabV3+, although it had a lower correction score. As shown in Figure 8d), this model introduced some white objects (that might have influenced the connection score) that should not belong to the final predicted image. In images 4 and 5 of Figure 8, the Unet model had some difficulties extracting complete roads on environments showing occlusions cast by trees and shadows, producing incomplete road sections.

Lastly, the FPN model generated complete roads with sharp edges, overcoming complex backgrounds even with multiple trees covering the area. The FPN model also generated some white objects from incorrectly predicted labeling that should be later removed.

DeepLabV3+, Unet, and FPN Network models were measured with different metrics in the experiments. Figure 9 shows the average metrics of the rural road detection for the test dataset. The blue color represents *completeness*, the red color represents *correctness*, the yellow color represents *quality*, and the green color represents the F1 metric. All these metrics have values between 0 and 1.

Comparing the three models: (i) DeepLabV3+ achieved the highest average correctness score of the tree models; (ii) Unet achieved a lower score compared to DeepLabV3+; and (iii) FPN, does not outperform any of the highest scores; and FPN achieved the highest scores on average *completeness*, *quality*, and F1. Based on these results, DeepLabV3+ performs better if the priority is a high degree of correctness, and the FPN performs better if the goal is to have a higher degree of *completeness*, *quality*, and F1. In this research work, correctness was favored against completeness as the aim was to achieve a lower proportion of false positives (objects in the image wrongly detected as roads). The results are listed in Table 1.

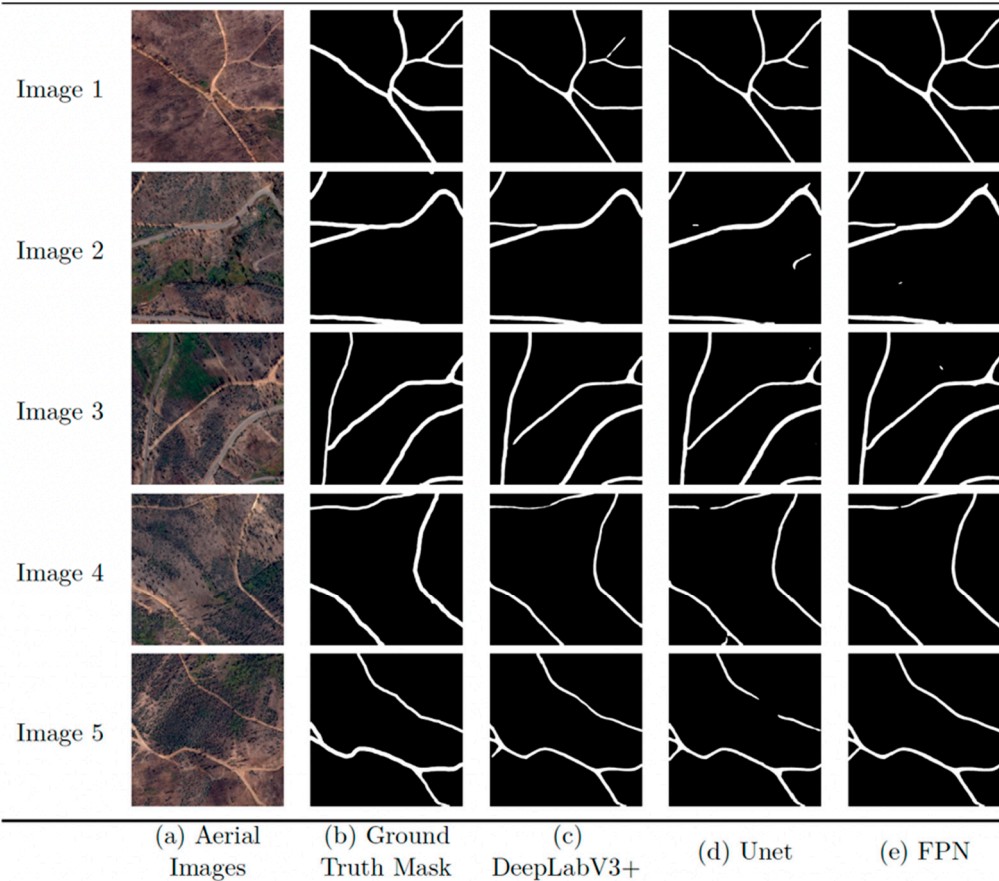

**Figure 8.** Rural road detection results over five sample input image tiles: (**a**) aerial images; (**b**) manually made ground truth masks; (**c**) images predicted by the DeepLabV3+ architecture; (**d**) images predicted by the Unet architecture; (**e**) images predicted by the FPN architecture.

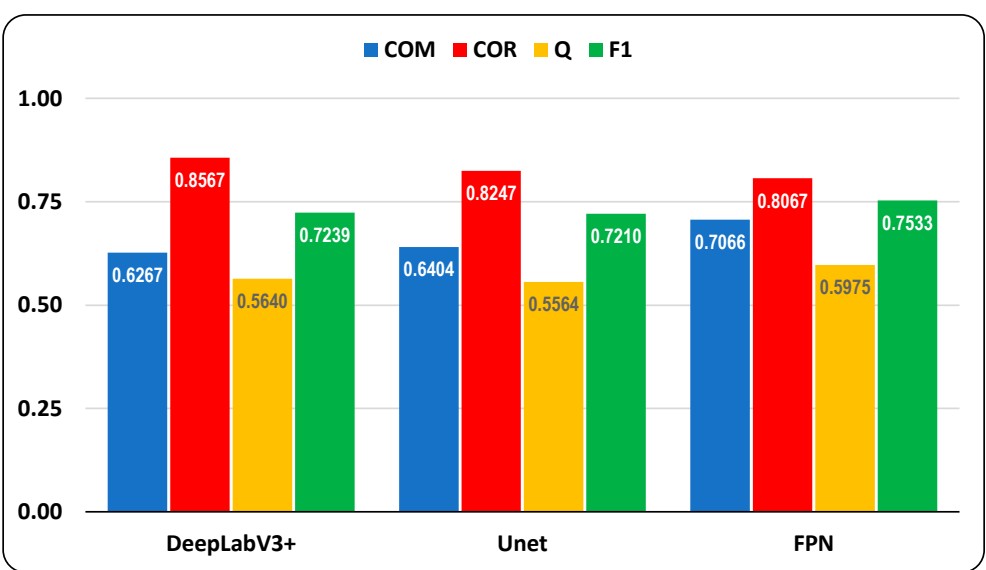

**Figure 9.** Rural road detection average metrics evaluation for the test dataset.

Figure 10(a1,b1) show an example of the DeepLabV3+ model's robustness against strong shadows and small bushes covering the road area. Here the model overcame a complex landscape situation and correctly predicted the rural road. With the existence of narrow rural roads and a high density of shadows and small bushes in the middle of the

road, it is visible in Figure 10(a2,b2) that the DeepLabV3+ model had no problems making relatively accurate predictions.

**Table 1.** Road region detection quantitative results.

| Architecture | Image 1 (of Figure 9) | | | | Image 2 (of Figure 9) | | | | Avg. (Test Set) | | | |
|---|---|---|---|---|---|---|---|---|---|---|---|---|
| | COM | COR | Q | F1 | COM | COR | Q | F1 | COM | COR | Q | F1 |
| DeepLabV3+ | 0.6368 | 0.9573 | 0.6192 | 0.7648 | 0.8380 | 0.9353 | 0.7921 | 0.8840 | 0.6267 | 0.8567 | 0.5640 | 0.7239 |
| Unet | 0.6767 | 0.9464 | 0.6517 | 0.7891 | 0.7519 | 0.8733 | 0.6780 | 0.8081 | 0.6404 | 0.8247 | 0.5564 | 0.7210 |
| FPN | 0.7739 | 0.9234 | 0.7272 | 0.8420 | 0.8518 | 0.9021 | 0.7798 | 0.8763 | 0.7066 | 0.8067 | 0.5975 | 0.7533 |

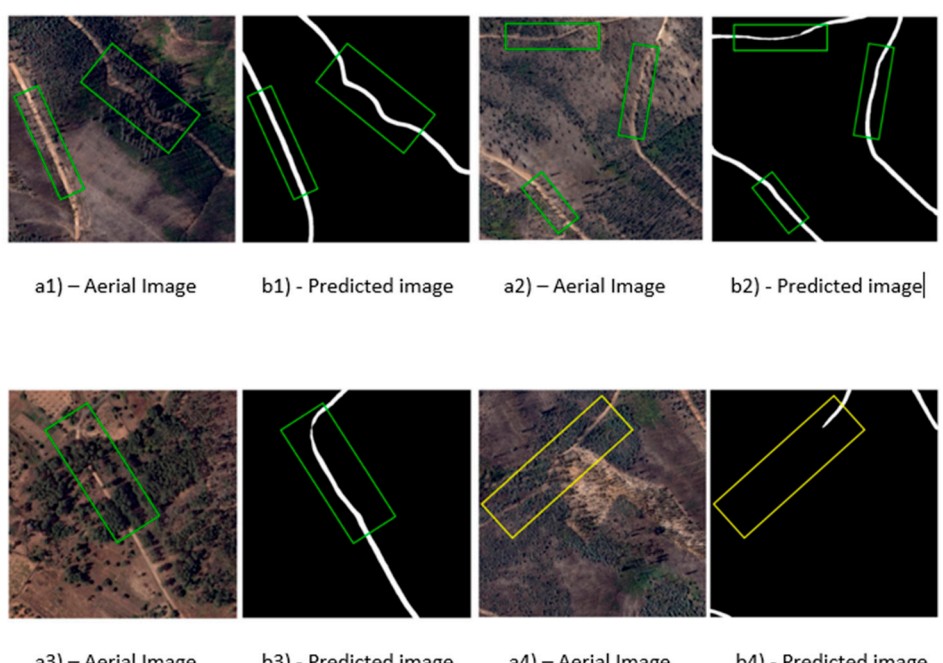

**Figure 10.** Sample results of rural road detection: (**a1**,**a2**,**a3**,**a4**) multiple aerial images with different road occlusions; (**b1**,**b2**,**b3**,**b4**) the predicted roads by the proposed method.

In Figure 10(a3,b3), despite having two road occlusions, the DeepLabV3+ model showed very high robustness even with the road completely obstructed by trees and vegetation. The only part where it misses a connection is near the top of the green rectangle, where the road changed from asphalt to dirt.

Finally, Figure 10(a4,b4) present a case where the DeepLabV3+ model was not able to detect the rural roads properly due to a highly complex scenario, exhibiting different road textures, severe shadows, and other terrain patterns not particularly well-defined. In these cases, it can be challenging even for a human operator to decide with certainty whether a rural road exists.

*5.3. Road Centerline Extraction Evaluation*

A comparison between two morphological thinning methods was made for road centerline extraction. The first was the Zhang–Suen algorithm [22], while the second was the Guo–Hall algorithm [23]. The optimization process was applied before the two thinning algorithms. Their results will be thoroughly described in the following section.

5.3.1. Road Centerline Extraction with Zhang–Suen Algorithm

Sample results of the Zhang–Suen Thinning algorithm applied to different models are presented in Figure 11. Visual analysis shows that DeepLabV3+ produces almost correct and smooth roads, although it failed on thinner roads, as in Image 1, and in road connectivity, as in Image 3. The Unet in Image 1 completed the thinner lines and road

connections but introduced some FPs in Image 2 (white lines on the exterior of the reference map). It is noticeable that Unet completed the connections on Image 3 but performed poorly on Images 4 and 5, leaving FNs (pixels wrongly identified as background) and incomplete road branches in places with high forest density covering the road. The FPN achieved a complete detection of rural roads only with some extra FPs. Sometimes, identifying these FPs is challenging even by a human operator because due to the image background's complexity.

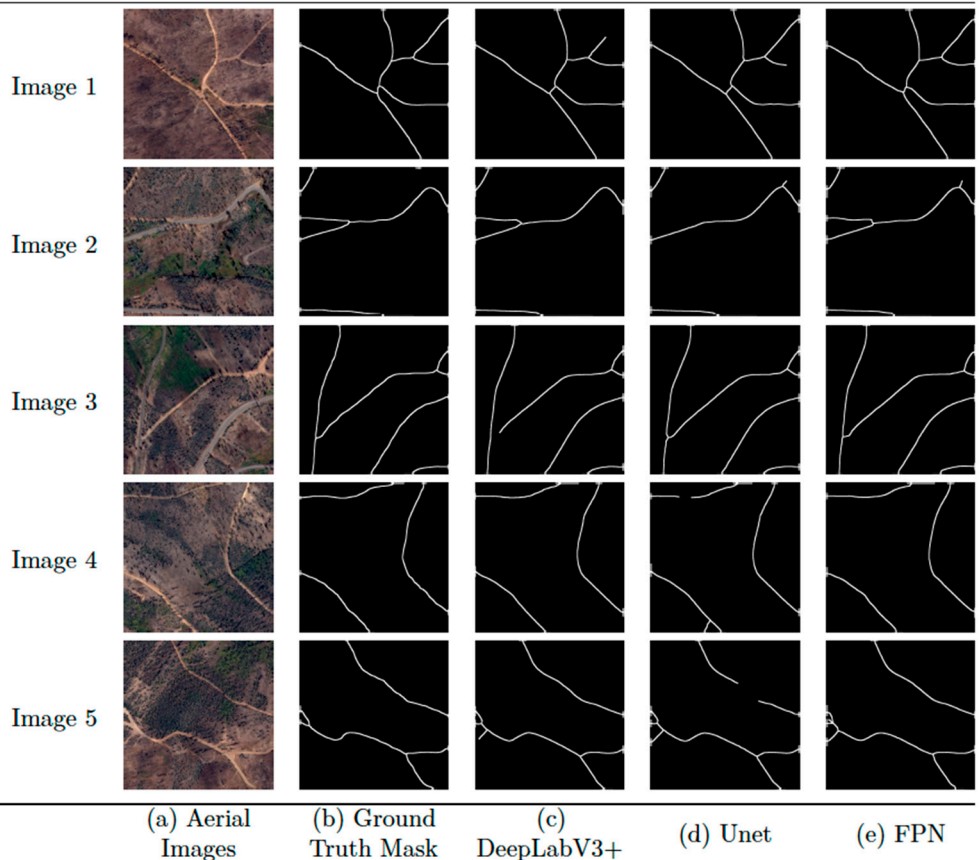

|  | (a) Aerial Images | (b) Ground Truth Mask | (c) DeepLabV3+ | (d) Unet | (e) FPN |

**Figure 11.** Rural road extraction network for the Mação district using the Zhang–Suen thinning algorithm—a comparison between multiple roads detected with different road detection algorithms studied.

Quantitative metrics are listed in Tables 2–5 for the chosen values of $\rho$ (corresponding to 0.5 m, 1 m, 1.5 m, and 2 m on the ground). After analyzing the average metrics obtained for the test dataset, it can be concluded that DeepLabV3+ outperformed Unet, and FPN models on *correctness*, *quality*, and F1 for the four different $\rho$-values. The Unet did not outperform any of the models on these metrics. The FPN model achieved the highest *completeness* score of the three models. From these experiments, it can be observed that DeepLabV3+ presented the roads correctly and FPN the complete ones.

**Table 2.** Rural road extraction quantitative results for a $\rho = 2$ (using the Zhang–Suen thinning algorithm)—values in bold represent the road detection algorithm with the best performance in each parameter.

| $\rho = 2 = 0.5$ m | Image 1 | | | | Image 2 | | | | Avg. (Test Set) | | | |
|---|---|---|---|---|---|---|---|---|---|---|---|---|
| Architecture | COM | COR | Q | F1 | COM | COR | Q | F1 | COM | COR | Q | F1 |
| DeepLabV3+ | **0.5539** | **0.5933** | **0.4015** | **0.5729** | 0.6583 | 0.6403 | 0.4806 | 0.6492 | 0.4758 | **0.4850** | **0.3184** | **0.4804** |
| Unet | 0.5377 | 0.5646 | 0.3801 | 0.5508 | 0.5918 | 0.6199 | 0.4342 | 0.6055 | 0.4681 | 0.4634 | 0.3052 | 0.4657 |
| FPN | 0.5067 | 0.5071 | 0.3395 | 0.5069 | **0.6761** | **0.6597** | **0.5013** | **0.6678** | **0.4894** | 0.4646 | 0.3164 | 0.4767 |

**Table 3.** Rural road extraction quantitative results for a $\rho = 4$ (using the Zhang–Suen thinning algorithm)—values in bold represent the road detection algorithm with the best performance in each parameter.

| $\rho = 4 = 1$ m | Image 1 | | | | Image 2 | | | | Avg. (Test Set) | | | |
|---|---|---|---|---|---|---|---|---|---|---|---|---|
| **Architecture** | **COM** | **COR** | **Q** | **F1** | **COM** | **COR** | **Q** | **F1** | **COM** | **COR** | **Q** | **F1** |
| DeepLabV3+ | 0.7599 | 0.8045 | 0.6415 | 0.7816 | **0.8376** | 0.8188 | 0.7066 | 0.8281 | 0.6937 | **0.7076** | **0.5441** | **0.7006** |
| Unet | 0.8026 | **0.8274** | 0.6875 | 0.8148 | 0.7263 | 0.7632 | 0.5927 | 0.7443 | 0.6840 | 0.6758 | 0.5178 | 0.6799 |
| FPN | **0.8268** | 0.8153 | **0.6964** | **0.8210** | 0.8366 | **0.8240** | **0.7098** | **0.8303** | **0.7109** | 0.6756 | 0.5359 | 0.6928 |

**Table 4.** Rural road extraction quantitative results for a $\rho = 6$ (using the Zhang–Suen thinning algorithm)—values in bold represent the road detection algorithm with the best performance in each parameter.

| $\rho = 6 = 1.5$ m | Image 1 | | | | Image 2 | | | | Avg. (Test Set) | | | |
|---|---|---|---|---|---|---|---|---|---|---|---|---|
| **Architecture** | **COM** | **COR** | **Q** | **F1** | **COM** | **COR** | **Q** | **F1** | **COM** | **COR** | **Q** | **F1** |
| DeepLabV3+ | 0.8480 | 0.8983 | 0.7738 | 0.8725 | **0.9227** | **0.9063** | **0.8423** | **0.9144** | 0.8063 | **0.8234** | **0.6940** | **0.8148** |
| Unet | 0.9181 | **0.9427** | 0.8696 | 0.9302 | 0.7898 | 0.8357 | 0.6834 | 0.8121 | 0.8013 | 0.7908 | 0.6621 | 0.7960 |
| FPN | **0.9592** | 0.9425 | **0.9062** | **0.9508** | 0.9009 | 0.8904 | 0.8110 | 0.8957 | **0.8187** | 0.7796 | 0.6695 | 0.7987 |

**Table 5.** Rural road extraction quantitative results for a $\rho = 8$ (using the Zhang–Suen thinning algorithm)—values in bold represent the road detection algorithm with the best performance in each parameter.

| $\rho = 8 = 2$ m | Image 1 | | | | Image 2 | | | | Avg. (Test Set) | | | |
|---|---|---|---|---|---|---|---|---|---|---|---|---|
| **Architecture** | **COM** | **COR** | **Q** | **F1** | **COM** | **COR** | **Q** | **F1** | **COM** | **COR** | **Q** | **F1** |
| DeepLabV3+ | 0.9211 | 0.9749 | 0.8998 | 0.9473 | **0.9368** | **0.9240** | **0.8698** | **0.9304** | 0.8489 | **0.8683** | **0.7588** | **0.8585** |
| Unet | 0.9605 | **0.9826** | 0.9444 | 0.9714 | 0.8031 | 0.8505 | 0.7037 | 0.8261 | 0.8524 | 0.8434 | 0.7363 | 0.8479 |
| FPN | **0.9938** | 0.9805 | **0.9746** | **0.9871** | 0.9203 | 0.9137 | 0.8468 | 0.9170 | **0.8709** | 0.8308 | 0.7430 | 0.8504 |

### 5.3.2. Road Extraction with Guo–Hall Algorithm

In Figure 12, the results of the road extraction using the Guo–Hall thinning algorithm are shown. On this test, the results were very similar to those obtained using the Zhang–Suen thinning algorithm, not showing significant discrepancies between different $\rho$-values (see Tables 6–9) and achieving very similar results (a difference of less than 1% between them). Once again, DeepLabV3+ achieved superior results on *correctness*, *quality*, and F1. The Unet did not outperform any of the three models, and FPN again achieved the highest *completeness* score.

**Table 6.** Rural road extraction quantitative results for a $\rho = 2$ (using the Guo–Hall thinning algorithm).

| $\rho = 2 = 0.5$ m | Image 1 | | | | Image 2 | | | | Avg. (Test Set) | | | |
|---|---|---|---|---|---|---|---|---|---|---|---|---|
| **Architecture** | **COM** | **COR** | **Q** | **F1** | **COM** | **COR** | **Q** | **F1** | **COM** | **COR** | **Q** | **F1** |
| DeepLabV3+ | **0.5336** | **0.5587** | **0.3754** | **0.5459** | 0.6205 | 0.5969 | 0.4373 | 0.6085 | 0.4760 | **0.4801** | **0.3167** | **0.4780** |
| Unet | 0.5118 | 0.5274 | 0.3509 | 0.5195 | 0.5546 | **0.6307** | 0.4187 | 0.5902 | 0.4661 | 0.4621 | 0.3034 | 0.4641 |
| FPN | 0.4882 | 0.4777 | 0.3183 | 0.4829 | **0.6280** | 0.6167 | **0.4517** | **0.6223** | **0.4925** | 0.4639 | 0.3163 | 0.4778 |

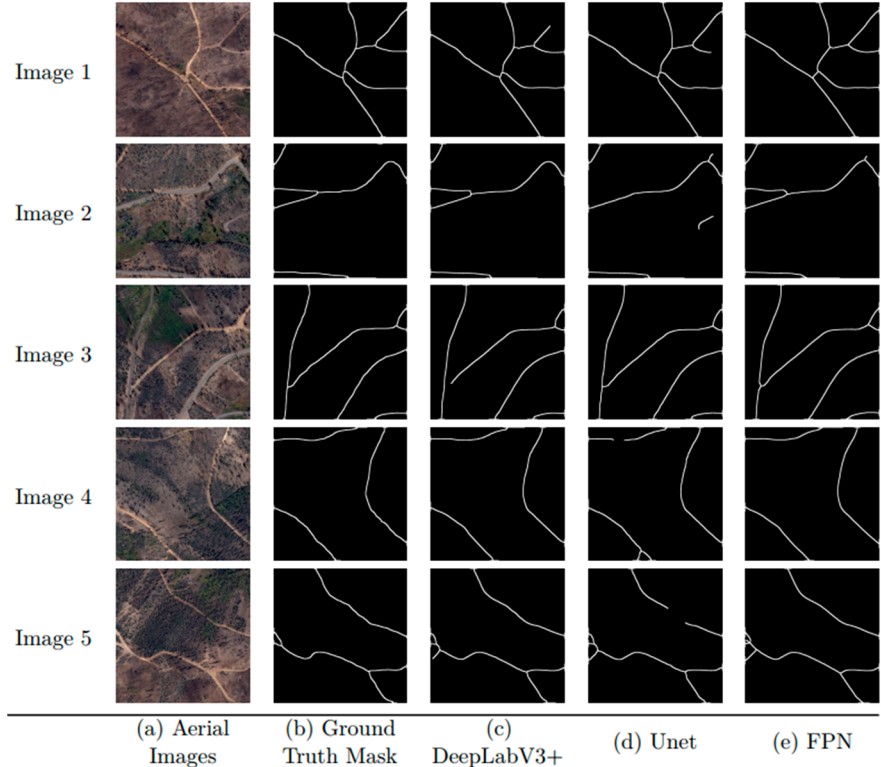

**Figure 12.** Rural road extraction network for the Mação district using the Guo Hall algorithm to perform the thinning—a comparison between multiple roads detected with different road detection algorithms studied.

**Table 7.** Rural road extraction quantitative results for a $\rho = 4$ (using the Guo–Hall thinning algorithm).

| $\rho = 4 = 1$ m | Image 1 | | | | Image 2 | | | | Avg. (Test Set) | | | |
|---|---|---|---|---|---|---|---|---|---|---|---|---|
| Architecture | COM | COR | Q | F1 | COM | COR | Q | F1 | COM | COR | Q | F1 |
| DeepLabV3+ | 0.7506 | 0.7857 | 0.6231 | 0.7678 | 0.8253 | 0.8004 | 0.6845 | 0.8127 | 0.6947 | 0.7047 | 0.5428 | 0.6997 |
| Unet | 0.7799 | 0.8017 | 0.6538 | 0.7907 | 0.7129 | 0.8098 | 0.6106 | 0.7583 | 0.6831 | 0.6785 | 0.5171 | 0.6808 |
| FPN | 0.8170 | 0.7984 | 0.6772 | 0.8076 | 0.8245 | 0.8130 | 0.6930 | 0.8187 | 0.7135 | 0.6748 | 0.5351 | 0.6936 |

**Table 8.** Rural road extraction quantitative results for a $\rho = 6$ (using the Guo–Hall thinning algorithm).

| $\rho = 6 = 1.5$ m | Image 1 | | | | Image 2 | | | | Avg. (Test Set) | | | |
|---|---|---|---|---|---|---|---|---|---|---|---|---|
| Architecture | COM | COR | Q | F1 | COM | COR | Q | F1 | COM | COR | Q | F1 |
| DeepLabV3+ | 0.8475 | 0.8905 | 0.7675 | 0.8685 | **0.9179** | **0.8930** | **0.8269** | **0.9053** | 0.8037 | **0.8176** | **0.6876** | **0.8106** |
| Unet | 0.9220 | **0.9498** | 0.8792 | 0.9357 | 0.7763 | 0.8830 | 0.7039 | 0.8262 | 0.8020 | 0.7980 | 0.6662 | 0.8000 |
| FPN | **0.9691** | 0.9476 | **0.9198** | **0.9582** | 0.9002 | 0.8833 | 0.8045 | 0.8917 | **0.8201** | 0.7796 | 0.6697 | 0.7993 |

**Table 9.** Rural road extraction quantitative results for a $\rho = 8$ (using the Guo–Hall thinning algorithm).

| $\rho = 8 = 2$ m | Image 1 | | | | Image 2 | | | | Avg. (Test Set) | | | |
|---|---|---|---|---|---|---|---|---|---|---|---|---|
| Architecture | COM | COR | Q | F1 | COM | COR | Q | F1 | COM | COR | Q | F1 |
| DeepLabV3+ | 0.9379 | 0.9808 | 0.921 | 0.9589 | **0.9409** | **0.9191** | **0.869** | **0.9299** | 0.8492 | **0.8659** | **0.7568** | **0.8575** |
| Unet | 0.9586 | **0.9882** | 0.9477 | 0.9732 | 0.7921 | 0.9062 | 0.7321 | 0.8453 | 0.8520 | 0.8518 | 0.7417 | 0.8519 |
| FPN | **0.9959** | 0.9813 | **0.9773** | **0.9885** | 0.9228 | 0.9113 | 0.8467 | 0.9170 | **0.8719** | 0.8309 | 0.7432 | 0.8509 |

The Guo–Hall thinning algorithm was also tested to extract rural road centerlines. Figure 13 shows a visual comparison between the Zhang–Suen and Guo–Hall thinning algorithms. The Guo–Hall, when compared with the Zhang–Suen algorithm, produces more curved lines (as seen inside the red rectangles). This effect is discernible in road interceptions.

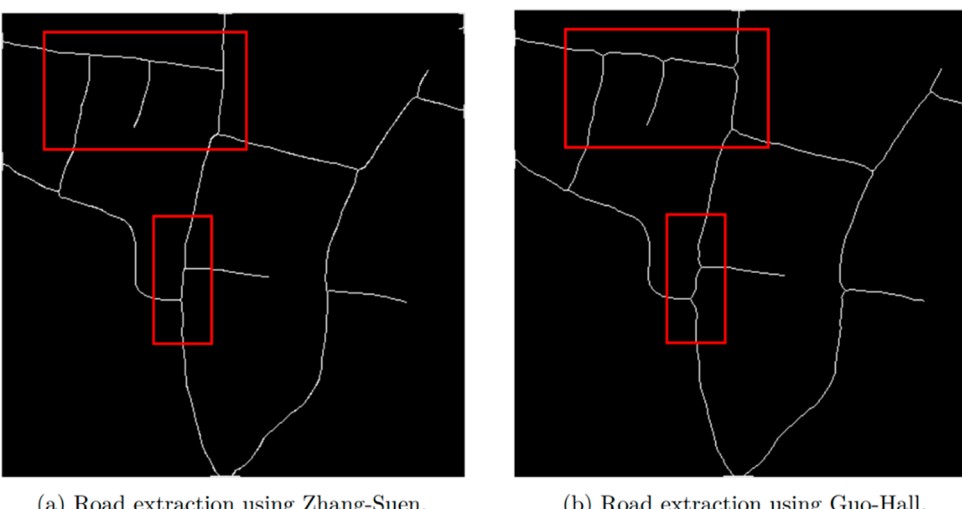

(a) Road extraction using Zhang-Suen.     (b) Road extraction using Guo-Hall.

**Figure 13.** A sample comparison between the Zhang–Suen and the Guo–Hall thinning algorithms.

## 6. Discussion

From the above experimental results, the best configuration obtained with the DeepLabV3+ model was with a buffer width of $\rho = 8$ (corresponding to a width of 2 m on the field) using the Zhang–Suen thinning algorithm to extract the rural roads. The following metrics were achieved: *completeness* = 0.8489, *correctness* = 0.8683, *quality* = 0.7588, and F1 = 0.8585.

When comparing *quality* with the method developed by [12], the proposed method obtained a lower score, but on the other side, achieved superior results on road centerline extraction concerning their Huang-C and Miao-C method for $\rho = 1$ and $\rho = 2$. In this research, the authors claim for $\rho = 1$, Huang-C *quality* = 0.6471, Miao-C *quality* = 0.6218, and for $\rho = 2$, Huang-C *quality* = 0.7027, Miao-C *quality* = 0.6735. It should also be noted that in Cheng et al. work, each $\rho$ pixel incrementation corresponds to 1.2 m per pixel, which means that the buffer width for $\rho = 2$ translates to a size of 2.4 m, which is a wider buffer when compared to our proposed method.

Also, our method achieved a higher *quality* when compared with Zhang et al. [14] for the Guangzhou dataset achieving higher scores than Huang and Miao's method. Their quality score = 0.7522, their Huang's method quality = 0.6890, their Miao's method quality = 0.7169.

The experiments were not tested with the same variables and dataset, but overall, the results achieved are very auspicious.

## 7. Conclusions

During the last decades, Portugal has suffered from extreme forest fires that can be substantially mitigated with the use of recent technology. Aiming to preserve forests, reduce the danger to society, and help firefighters to have higher efficiency and increased situational awareness in the field, a method to automatically detect and extract rural road centerlines from aerial images was proposed to enable car navigation on roads typically unavailable on typical navigation applications. The goal was to create a faster and more precise process than the previous handmade tedious work undertaken by aerial image visual interpreters.

The proposed method uses and compares recent deep learning methodologies like DeepLabV3+, Unet, and FPN models to detect rural roads. Later it uses morphological algorithms to optimize the connections between different types of roads, ending with thinning algorithms like Zhang–Suen and Guo–Hall to extract the rural road centerlines.

For assessing the performance of the presented architecture, four metrics were calculated: *completeness*, *correctness*, *quality*, and F1. Although, all metrics were important for the performance evaluation, *completeness* and *correctness*, shown to be more informative select the road detection algorithms.

For the rural road centerline extraction, the top performance was achieved by the proposed method with DeepLabV3+ as the road detector, combined with a buffer width of $\rho = 8$ (corresponding to a width of 2 m on the field), and using the Zhang–Suen thinning algorithm to extract the rural roads. This configuration brought the best metrics achieving a *completeness* = 0.8489, *correctness* = 0.8683, *quality* = 0.7588, and F1 = 0.8585.

Although the proposed method was the best performing on road centerline extraction phase, it was noticed that the FPN model achieved the highest F1 score for road detection and the highest *completeness* score on all the road centerline extraction tests. If the priority for the end-user is only to detect rural roads or extract rural road centerlines with a higher degree of *completeness*, the use of the FPN model should also be considered.

The proposed method provides a very efficient and feasible solution for accurate rural road detection and centerline extraction from aerial images. It surpasses strong shadows by trees, small bushes, and vegetation in the middle of the road. With the optimization, it is also possible to connect different types of roads and overcome total road occlusions with a considerable size.

*Future Work*

As future work, further improvements should focus on the following topics:

- Validation of the results on the terrain. This task will confirm that the methods developed in this research work are accurate and verified in the field to create future rural road networks;
- Adding the rural roads' centerline data into a geographical information system, followed by the incorporation of the data into real-time mapping applications;
- Increasing the size and quality of the training dataset, becoming more precise, focusing on what pixels, belong to the background and what pixels belong to the rural roads. The more samples the datasets have, the more scenarios and complex backgrounds the models can learn from and thus become better at making predictions, reducing possible situations of overfitting;
- Creating an algorithm specifically to detect incomplete road intersections and dead-end roads;
- It would be relevant to start utilizing RGB images with the infrared component, so the deep learning models can also learn about the reflectance of green zones with vegetation and trees;
- Run the algorithm as a web service so that the user can query it with his location to obtain the entire road network (including rural roads) in a radius of $n$ km;
- Incorporate on the road network the information of which vehicles can use the different road segments according to the vehicle's dimensions. Allow the user to provide this information to the web service alongside his position.

**Author Contributions:** Conceptualization, M.L., D.E., L.O., H.O. and A.M.; methodology, M.L., D.E., L.O. and H.O.; software, M.L. and D.E.; validation, M.L., D.E., L.O., H.O. and A.M.; formal analysis, M.L., D.E., L.O., H.O. and A.M.; investigation, M.L., D.E., L.O., H.O. and A.M.; resources, L.O., H.O. and A.M.; data curation, M.L., D.E. and H.O.; writing—original draft preparation, M.L., D.E., L.O. and H.O.; writing—review and editing, M.L., D.E., L.O., H.O. and A.M.; supervision, L.O., H.O. and A.M.; project administration, L.O., H.O. and A.M.; funding acquisition, L.O., H.O. and A.M. All authors have read and agreed to the published version of the manuscript.

**Funding:** This work was financially supported by FCT (National Foundation of Science and Technology) within the Research Unit CTS—Centre of Technology and Systems, UIDB/00066/2020 and through Project foRESTER (PCIF/SSI/0102/2017, http://www.forester.pt, accessed on 19 October 2022) and project PEST (UIDB/00066/2020).

**Data Availability Statement:** All data and materials in this article can be obtained by contacting the corresponding author, Henrique Oliveira (hjmo@lx.it.pt).

**Acknowledgments:** We would like to thank the authorities from the Municipality of Mação, particularly to António Louro, for the valuable support in establishing the user requirements and the feedback for the system's validation.

**Conflicts of Interest:** The authors declare no conflict of interest.

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
