# Peer review of "Automatic Rural Road Centerline Detection and Extraction from Aerial Images for a Forest Fire Decision Support System"

_remotesensing, doi:10.3390/rs15010271_

Round 1

Reviewer 1 Report

This paper introduces a very efficient and feasible solution for accurate rural road detection and centerline extraction from aerial images referring to domain of interest at Portugal.  

The quality of the paper is technically sound. A standard data set is used for the analysis and testing of the method. The paper has illustrated  the performance evaluation and effectiveness of the proposed method, but needs a small revision as discussed below.

1.All keywords (abbreviations) must be mentioned in the abstract.

2. A diagram is expected to understand the proposed road detection architecture rather than sequence of statements as discussed in section 4.2

3.Pseudo code at section 4.3,4.4 is expected to understand the overall methodology in specific.

4. A comparison of the various qualitative and quantitative methods as used in  Section 5.2 in a tabular format is expected to better understand the road evaluation techniques.

Author Response

The reply to reviewers is on the attached file.

Reviewer 2 Report

The paper addresses rural roads detection using deep learning. This is an interesting and relevant topic that will probably reduce the time to mitigate a forest fire. The paper is grammatically well written, making it easy to read. However there are some scientific issues, as follow:

- Sections 1 and 2 should be rewritten in order to focus in the rural roads detection instead of different aspects regarding with dealing with fires.

* In those sections, the authors' contributions must be clearly highlighted. The contributions are not presented, neither any detail regarding the methodological approach. It is not clear whether it is proposed a new algorithm or just a methodology using algorithms previously proposed in the literature.

* In Section 1 only two lines (80,81) present the paper objectives. 

* Section 2 looks like an introduction. Only the last paragraph (lines 128-138) BRIEFLY discusses other approaches to road detection and centerline extraction. This section should focus in to review papers dealing with rural road centerline detection and extraction, not in techniques to prevent a fire.

- Sections 5, 6 and 7 are confused and should be rewritten in order to better describe the simulations, comparisons and conclusions.

* There are several methods, executed several times, with different configurations, yielding several comparisons not clearly presented.

Some questions:

- Line 143: The authors should explain the values 89.0% and 82.1%.  What are they referring to?

- Section 4: What if the dilation step connect roads that are not connected in the ground truth image? 

- The empirical size of 140 pixels for objectes that be removed is specific for this dataset, right? Did the authors test with different datasets? (Lines 376-378)

- How to distinguish between an oclusion in the image and a non-connected road?

- Figure 9: except by COR (only 5% of difference), FPN seems better. Did not this make FPN a more suitable choice than DeepLabV3+? Figure 11 exemplify this hypotesis.

- What is the difference between tables 1-4 and 5-8? Both groups shows Rural road extraction quantitative results for \rho = {2,4,6,8}. Does one group use Zhang-Suen Algorithm, while the other uses Guo-Hall Algorithm? This is not said in the paper.

- Lines 534-536: "there are two metrics that (...) should consider (...): completeness and 535 correctness". What about quality and F1? Why do the authors address these other metrics, if they are not considered?

Other comments:

- Quality of Figures. Several of them looks like to be snatshooted from a PDF.

- The authors could improve the literature review by reading and citing the papers [1] and [2]. In [1] it is presented a discussion regarding visual quality, including the impact of occlusion in monitoring areas, as mentioned by the authors. In [2] it is discussed several methods to deal with emergencies in a smart scenario, specially the leverage of using artificial intelligence in those situations.

[1] Jesus, T.C. et al., F. "A Survey on Monitoring Quality Assessment for Wireless Visual Sensor Networks". Future Internet 2022, 14, 213.

[2] Elvas, L.B.; Mataloto, B.M.; Martins, A.L.; Ferreira, J.C. Disaster Management in Smart Cities. Smart Cities 20214, 819-839.

Author Response

(The authors gave the same response as above.)

Round 2

Reviewer 2 Report

I am satisfied!